# Individual Fairness In Strategic Classification

**Zhiqun Zuo**
zuo.167@osu.edu

**Mohammad Mahdi Khalili**
khalili.17@osu.edu

Department of Computer Science and Engineering
The Ohio State University
Columbus, OH 43210

## Abstract

Strategic classification, where individuals modify their features to influence machine learning (ML) decisions, presents critical fairness challenges. While group fairness in this setting has been widely studied, individual fairness remains underexplored. We analyze threshold-based classifiers and prove that deterministic thresholds violate individual fairness. Then, we investigate the possibility of using a randomized classifier to achieve individual fairness. We introduce conditions under which a randomized classifier ensures individual fairness and leverage these conditions to find an optimal and individually fair randomized classifier through a linear programming problem. Additionally, we demonstrate that our approach can be extended to group fairness notions. Experiments on real-world datasets confirm that our method effectively mitigates unfairness and improves the fairness-accuracy trade-off.

## 1 Introduction

As machine learning (ML) becomes increasingly popular in decision-making systems, the interaction between individuals and AI agents emerges as a critical issue. AI-aided applications, such as automated lending [1], resume screening [2], and college admissions [3], significantly influence people's opportunities and outcomes. A fundamental issue in these applications is fairness—ensuring that ML-based decisions do not systematically disadvantage certain individuals or groups. Various fairness notions, including tier balancing [4], lookahead counterfactual fairness [5], and long-term fairness [6], have been proposed to address fairness issues arising from interactions between human and AI.

A major challenge in ML-based decision-making process is strategic behavior—where individuals adjust their features to obtain more favorable outcomes [7, 8, 9]. Because ML models operate based on predefined decision criteria, they are inherently vulnerable to such gaming. For example, students may participate in specific extracurricular activities if they believe doing so will improve their college admissions prospects. The field of strategic classification seeks to address these challenges by designing classifiers that account for individuals' ability to adjust their features [7]. Prior work has examined diverse aspects of strategic classification, including heterogeneous strategic behavior [10], incentive-aware risk minimization [11], and performative prediction, where data distributions evolve in response to decision rules [12].

Fairness in ML has been widely studied, leading to the development of multiple fairness criteria such as statistical parity [13, 14], equal opportunity [15], counterfactual fairness [16, 17], and individual fairness [18]. However, fairness in strategic settings introduces new complexities. While some research has explored how strategic behavior may mitigate or exacerbate unfairness [19], much of the focus has been on group fairness [20, 21]. Less attention has been given to individual fairness even as strategic behavior amplifies disparities near decision thresholds.

39th Conference on Neural Information Processing Systems (NeurIPS 2025).

Beyond fairness in prediction outcomes, strategic classification raises concerns about fairness in social burden (i.e., the cost individuals must pay to meet decision criteria) [22]. Since different individuals face varying levels of difficulty in modifying their features, strategic decision-making can lead to unfair cost disparities. Prior work has examined the trade-off between social cost and institutional utility [22] and proposed group-specific thresholds to mitigate cost disparities [23]. However, ensuring individual fairness with respect to individual cost remains an open problem. For instance, two individuals with similar underlying qualifications may be forced to pay different costs to receive the same decision simply because one lies just below the decision threshold.

In this work, we analyze binary strategic classification and demonstrate that deterministic threshold-based classifiers inherently violate individual fairness. To address this, we propose a randomized thresholding approach, proving that individual fairness can be achieved if the threshold distribution satisfies certain conditions. Furthermore, we show that the optimal threshold distribution can be efficiently computed via linear programming. Finally, we investigate the challenge of simultaneously achieving individual and group fairness, proving that this can be accomplished by adding a constraint to the threshold distribution—while still preserving the linear programming formulation.

Our results provide a principled framework for fair decision-making in strategic classification, ensuring equitable treatment both in terms of predictions and the costs individuals must bear to adapt. By addressing fairness from both an individual and group perspective, our approach offers a practical and computationally efficient solution to mitigate unfairness in real-world strategic decision-making systems.

## 2 Notations and Preliminaries

### 2.1 Strategic Classification

In this paper, we consider a binary classification problem in a strategic environment. The goal is to find a classifier $f : \mathcal{X} \mapsto \mathcal{Y}$ to predict binary label $Y$ from the observed features $X$ with $X \in \mathcal{X}$ and $Y \in \mathcal{Y} = \{0, 1\}$. In a strategic setting, $f$ is being used by a decision maker/institution to identify qualified individuals for a job or position, and individuals can improve their features $x$ to get a better result.[1] When an individual adjust feature $x$ to $x'$, a cost will be incurred which is represented by cost function $c : \mathcal{X} \times \mathcal{X} \mapsto \mathbb{R}^+ \cup \{0\}$ with $c(x, x) = 0, \forall x \in \mathcal{X}$. The institution and individuals aim at maximizing their utility functions. The utility function of the institution is the accuracy on the improved/adjusted features, i.e., $\mathcal{U}_{inst}(f) = \Pr\{f(X') = Y\}$.

The individuals need to consider both the gain from the classification and the cost they need to pay for the feature improvement. So, their utility is defined as follows, $\mathcal{U}_{indiv}(f) = [f(x') - f(x)] - \lambda c(x, x')$, where $\lambda$ is a positive constant measuring the relative importance of the cost to people. The best response of a person is $x'$ such that $\mathcal{U}_{indiv}(f)$ is maximized. We denote the best response to classifier $f$ as $\Delta x(f) = \arg\max_{x'} \mathcal{U}_{indiv}(f)$.

In this paper, we will analyze the strategic classification problem where $x$ is a general $d$-dimensional feature vectors. The following definition and assumption will be used throughout this paper.

**Definition 2.1** ($d$-dimensional binary classification problem). *A $d$-dimensional binary classification problem is a problem of finding a classifier $f : \mathcal{X} \to \mathcal{Y}$ with $\mathcal{X} \subseteq \mathbb{R}^d$ and $\mathcal{Y} = \{0, 1\}$.*

**Assumption 2.1.** *In a $d$-dimensional binary classification problem, we assume there is a function $l(x) : \mathbb{R}^d \mapsto \mathbb{R}$, such that*

$$l(x_1) \geq l(x_2) \quad \textit{iff} \quad \Pr\{Y = 1 | X = x_1\} \geq \Pr\{Y = 1 | X = x_2\}, \tag{1}$$

*and*

$$l(x_1) \leq l(x_2) \quad \textit{iff} \quad \Pr\{Y = 1 | X = x_1\} \leq \Pr\{Y = 1 | X = x_2\}. \tag{2}$$

*We further assume that $l(x)$ is a continuous and differentiable function and $l(X) \in [C, D]$.*

Note that we adopt Assumption 2.1, as it is commonly used in the literature [22]. The existence of $l(x)$ ensures that the problem can be addressed using threshold-based classifiers on $l(x)$. In many real-world scenarios (for example, when a bank decides whether to issue a loan based on an applicant's credit score) $l(x)$ could simply be $x$, since a higher credit score typically indicates

---

[1]Capital letters are being used for random variables, and small letters are being used for realizations.

a lower risk of default. When $x$ lies in a high-dimensional space, a reasonable choice for $l(x)$ is $l(x) = \Pr\{Y = 1 \mid X = x\}$.

In this paper, we focus on **threshold classifiers**. A deterministic threshold classifier in a $d$-dimensional binary classification problem is given by,

$$f(x;t) = \mathbf{1}\left[l(\Delta x(f)) \geq t\right], \tag{3}$$

where $t$ is a fixed threshold associated with classifier $f(x;t)$, and $\mathbf{1}(\cdot)$ is an indicator function [24]. Note that $\Delta x$ is used in (3) because the classifier is applied to adjusted features rather than original features.

## 2.2 Randomized Classifier

We denote a randomized classifier $\mathscr{F}$ as a random mapping from $\mathcal{X}$ to $\mathcal{Y}$. Each randomized classifier is associated with a function family $\mathcal{F}$ and a probability distribution $P$ over $\mathcal{F}$. Given $x$, the output distribution is,

$$\Pr\{\mathscr{F}(x) = y\} = \sum_{f \in \{h | h(x) = y, h \in \mathcal{F}\}} P(f). \tag{4}$$

A randomized threshold classifier is associated with a distribution over $\mathcal{F} = \{f(x;t) | t \in \mathcal{T}\}$, where $\mathcal{T}$ is the set of possible thresholds. Let $p(t)$ be a probability density function over $\mathcal{T}$. Then, for $\mathscr{F}$ associated with $p(t)$, we have,

$$\Pr\{\mathscr{F}(x) = y\} = \int_{t \in [\mathcal{T} \cap \{\tau | f(x;\tau) = y\}]} p(t)\mathrm{d}t. \tag{5}$$

Note that deterministic classifier $f(x;t_0)$ can be regarded as a special randomized classifier with $p(t) = \delta(t - t_0)$, where $\delta(\cdot)$ is the unit impulse function (also referred to as Dirac delta function [25]). For the rest of the paper, we are focusing on randomized and deterministic threshold classifiers.

## 2.3 Best Response Cost

Since in this paper we consider a strategic setting, the individuals always choose the best response to the classifier announced by the institution/decision maker. We define the Best Response Cost (BRC) associated with an individual with feature $x$ as follows, $c_f(x) = c(x, \Delta x(f))$.

For a randomized classifier $\mathscr{F}$, BRC is

$$c_{\mathscr{F}}(x) = \sum_{f \in \mathcal{F}} c(x, \Delta x(f))P(f) = \mathbb{E}\{c(x, \Delta x(\mathscr{F}))\}. \tag{6}$$

We assume that cost function $c(x, x')$ depends on the distance between $l(x)$ and $l(x')$. More precisely, we consider the following cost function,

$$c(x, x') = \begin{cases} g(l(x') - l(x)) & l(x') \geq l(x), \\ \infty & l(x') < l(x). \end{cases} \tag{7}$$

where $g : \mathbb{R} \mapsto \mathbb{R}$ is differentiable and is a strictly increasing function satisfying $g(0) = 0$. When $l(x) < l(x')$, the cost is infinite because we are considering a threshold-based classifier, and decreasing $l(x)$ will decrease the probability of being positively classified. Therefore, we have the following best response function,

$$l(\Delta(f(x;t))) = \begin{cases} l(x) & l(x) < \left[t - g^{-1}(\frac{1}{\lambda})\right] \text{ or } l(x) \geq t, \\ t & \left[t - g^{-1}(\frac{1}{\lambda})\right] \leq l(x) < t. \end{cases} \tag{8}$$

For notational convenience, we denote the constant $g^{-1}(\frac{1}{\lambda})$ as $\mathcal{C}$. When $l(x) < t - \mathcal{C}$, the individual cannot benefit from increasing $l(x)$ due to the high cost. When $l(x) \geq t$, the individual does not need to change the feature to change the outcome. Only when $t - \mathcal{C} \leq l(x) < t$, increasing $l(x)$ to $t$ will result in positive utility $\mathscr{U}_{indiv}(f)$.

## 2.4 Individual Fairness

Dwork *et al.* [26] proposes the notion of individual fairness that requires people with similar features to be treated similarly. We can adopt a similar definition for a strategic setting. Mathematically, individual fairness with respect to (w.r.t.) the expectation of outcome/decision $\mathscr{F}$ in a strategic setting is defined as follows,

$$|\mathbb{E}\{\mathscr{F}(x_1)\} - \mathbb{E}\{\mathscr{F}(x_2)\}| \leq M_p \|x_1 - x_2\|_2, \tag{9}$$

where $M_p$ is a predefined constant, $\|x_1 - x_2\|$ is the Euclidean distance between $x_1$ and $x_2$. We can also adopt individual fairness with respect to BRC. In particular, we say the individual fairness with respect to BRC is satisfied if the following holds,

$$|c_{\mathscr{F}}(x_1) - c_{\mathscr{F}}(x_2)| \leq M_c \|x_1 - x_2\|_2. \tag{10}$$

# 3 Individual Fairness with Repspect to BRC

In this section, we show for certain $l(x)$, a classifier with a deterministic threshold can never satisfy individual fairness with respect to BRC. Then, we identify conditions under which a randomized classifier satisfies individual fairness. We also discuss how to find a fair optimal randomized classifier.

Consider a deterministic classifier $f(x; t_0)$. Note that if $t_0 \leq C + \mathcal{C}$, then all the individuals after choosing the best response will be assigned label 1, and $f(x; t_0)$ cannot distinguish between qualified and unqualified individuals. Therefore, in the following theorem, we focus in case where $t_0 \in (C + \mathcal{C}, D)$.

**Theorem 3.1.** *Consider a d-dimensional binary classification problem with deterministic classifier $f(x; t_0)$ with $t_0 \in (C + \mathcal{C}, D)$. If $l(x)$ is reverse Lipschitz continuous, i.e.*

$$|l(x_1) - l(x_2)| \geq L_l \|x_1 - x_2\|_2, L_l < \infty, \forall x_1, x_2, \tag{11}$$

*then for any constant $M_c$, there exist $x_1, x_2 \in \mathcal{X}$ such that $|c_f(x_1) - c_f(x_2)| > M_c \|x_1 - x_2\|_2$.*

Condition (11) ensures that the change in $x$ can effectively change $l(x)$ to affect the classification result. From the proof of Theorem 3.1 in Appendix A.1, BRC is not continuous at $x_0$ where $l(x_0) = t_0 - \mathcal{C}$, and individual fairness does not hold. Therefore, we relax a deterministic threshold and consider a randomized threshold. The following theorem implies that we can achieve individual fairness with respect to BRC by adding constraints on distribution $p(t)$.

**Theorem 3.2.** *Consider a d-dimensional binary classification problem, and let $C_l = \max_{x \in \mathcal{X}} \|\nabla_x l(x)\|_2$, $C_g = \max_{l \in [0, \mathcal{C}]} g'(l)$, and $\mathcal{F} = \{f(x; t) | t \in (C + \mathcal{C}, D)\}$ for randomized classifier $\mathscr{F}$. If $p(t) \leq L_c$ with $L_c = \min\{\frac{\lambda M_c}{C_l}, \frac{M_c}{C_l C_g \mathcal{C}}\}$, then for constant $M_c$, $\forall x_1, x_2 \in \mathcal{X}$, we have,*

$$|c_{\mathscr{F}}(x_1) - c_{\mathscr{F}}(x_2)| \leq M_c \|x_1 - x_2\|_2. $$

Theorem 3.2 shows that by relaxing the hard threshold, the individuals with feature vector $x_0$ such that $l(x_0) = t_0 - \mathcal{C} - \epsilon$ ($\epsilon$ is a positive constant) now also have the opportunity to change the classification result by paying a certain cost.

## 3.1 Fair Optimal Randomized Classifier

Theorem 3.2 demonstrates the existence of randomized classifiers that can achieve individual fairness. However, not all threshold distributions yield high accuracy, so it is necessary to identify one that achieves the highest possible accuracy. In this section, our goal is to find a randomized classifier that minimizes the error rate while satisfying individual fairness. Let $e(t)$ denote the error rate of $f(\cdot; t)$. Then, we have,

$$e(t) = \int_C^{t-\mathcal{C}} \Pr\{Y = 1 | l(X) = l\} p_L(l) \mathrm{d}l + \int_{t-\mathcal{C}}^D \Pr\{Y = 0 | l(X) = l\} p_L(l) \mathrm{d}l, \tag{12}$$

where $p_L(l)$ is the probability density function of $L = l(X)$. Note that $\Pr\{l_1 \leq l(X) \leq l_2\} = \int_{l_1}^{l_2} p_L(l) \mathrm{d}l$. For notational convenience, we denote $\rho_0(l) = \Pr\{Y = 0 | l(X) = l\} p_L(l)$ and

$\rho_1(l) = \Pr\{Y = 1 | l(X) = l\}p_L(l)$. Hence, to find the optimal classifier $\mathscr{F}$, we need to solve the following optimization problem,

$$p_c(t) = \arg\min_{p(t)} \mathbb{E}_{t \sim p(t)}[e(t)] = \arg\min_{p(t)} \int_{C+\mathcal{C}}^{D} e(t)p(t)\mathrm{d}t$$

$$s.t. \int_{C+\mathcal{C}}^{D} p(t)\mathrm{d}t = 1, \ 0 \le p(t) \le L_c. \tag{13}$$

In the above optimization problem, $L_c$ is a constant that controls the fairness level. Smaller $L_c$ leads to a better fairness level. Solving problem 13 is difficult because $p(t)$ is a general function, and convex or linear optimization techniques cannot be directly used. To tackle the challenge, we consider an approximated solution where $p_c(t)$ is a *piecewise constant function*. Given a hyper-parameter $K$, we assume that $(C + \mathcal{C}, D)$ can be divided into $K$ bins with the $k$-th bin corresponding to $(C + \mathcal{C} + \frac{(k-1)(D-C-\mathcal{C})}{K}, C + \mathcal{C} + \frac{k(D-C-\mathcal{C})}{K}]$ and $p_c(t)$ has a constant value in each bin. We denote these constant values as $\{p_{c1}, p_{c2}, ..., p_{cK}\}$, and for simplicity, we denote the start point of the $k$-th bin as $s_k$, i.e. $s_k = C + \mathcal{C} + \frac{(k-1)(D-C-\mathcal{C})2}{K}$. Then, the average error rate for a randomized classifier is given by,

$$\int_{C+\mathcal{C}}^{D} e(t)p(t)\mathrm{d}t = \sum_{k=1}^{K} p_{ck} \int_{s_k}^{s_{k+1}} \left( \int_{C}^{t-\mathcal{C}} \rho_1(l)\mathrm{d}l + \int_{t-\mathcal{C}}^{D} \rho_0(l)\mathrm{d}l \right) \mathrm{d}t. \tag{14}$$

Define $A_{ck}$ as

$$A_{ck} = \int_{s_k}^{s_{k+1}} \left( \int_{C}^{t-\mathcal{C}} \rho_1(l)\mathrm{d}l + \int_{t-\mathcal{C}}^{D} \rho_0(l)\mathrm{d}l \right) \mathrm{d}t. \tag{15}$$

Because $A_{ck}$ is a constant and determined solely by the distribution of $l(X)$, the optimization problem 13 can be translated as the following linear optimization problem,

$$\min \sum_{k=1}^{K} A_{ck} p_{ck}, \quad s.t. \quad \sum_{k=1}^{K} \frac{p_{ck}(D - C - \mathcal{C})}{K} = 1, \ \ 0 \le p_{ck} \le L_c. \tag{16}$$

The above problem can be efficiently solved using linear programming techniques (e.g., simplex method [27]). Algorithm 1 and 2 shows how to find the optimal randomized classifier and use it to make a prediction.

---

**Algorithm 1** Finding optimal randomized classifier satisfying individual fairness w.r.t. BRC

---

**Input:** Training data $\mathcal{D} = \{x_i, y_i\}_{i=1}^{N}$, $\lambda$, $M_c$, function $g, l$
1: $\mathcal{C} \leftarrow g^{-1}(\frac{1}{\lambda})$, $L_c \leftarrow \min\{\frac{\lambda M_c}{C_l}, \frac{M_c}{C_l C_g \mathcal{C}}\}$
2: Estimate $A_{ck}$ from data with Eq. 15
3: Solve the following optimization problem,

$$p_{ck} = \arg\min \sum_{k=1}^{K} A_{ck} p_{ck}, \quad s.t. \quad \sum_{k=1}^{K} \frac{p_{ck}(D - C - \mathcal{C})}{K} = 1, \ \ 0 \le p_{ck} \le L_c$$

**Output:** $p_{ck}$

---

## 4 Individual Fairness with Respect to Outcome

In this section, we show that our proposed randomized classifier can also be used to improve individual fairness with respect to decision/outcome. Given a randomized classifier $\mathscr{F}$, we define the average prediction outcome as follows, $\hat{Y}_{\mathscr{F}}(x) = \sum_{f \in \mathcal{F}} f(x)P(f) = \mathbb{E}\{\mathscr{F}(x)\}$.

Individual fairness with respect to outcome with constant $M_p$ requires $|\hat{Y}_{\mathscr{F}}(x_1) - \hat{Y}_{\mathscr{F}}(x_2)| \le M_p||x_1 - x_2||_2, \forall x_1, x_2 \in \mathcal{X}$. For certain $l(x)$, we can prove any deterministic classifier cannot satisfy this constraint.

---

[2]We use $s_{K+1}$ to denote the end point of the last bin even though there is no actual $(K + 1)$-th bin.

---

**Algorithm 2** Inference with a randomized classifier

---

**Input:** Data point $x$, threshold distribution $p_{ck}$, intervals $(s_k, s_{k+1})$, $k \in \{1, ..., K\}$, function $l$

1: Sample $k$ from distribution $\Pr\{\mathcal{K} = k\} = \frac{p_{ck}}{\sum_{k=1}^{K} p_{ck}}$
2: $t \leftarrow s_k + \eta$, $\eta$ is a random noise sampling from a uniform distribution on $[0, s_{k+1} - s_k]$
3: $\hat{y} \leftarrow \mathbf{1}[l(x) \geq t]$

**Output:** $\hat{y}$

---

**Theorem 4.1.** *Consider a $d$-dimensional classification problem and a randomized classifier $\mathscr{F}$ with $p(t) = \delta(t_0)$ and $t_0 \in (C + \mathcal{C}, D)$. If $l(x)$ is reverse Lipschitz continuous, i.e.,*

$$|l(x_1) - l(x_2)| \geq L_l \|x_1 - x_2\|_2, L_l < \infty, \forall x_1, x_2, \tag{17}$$

*then for any constant $M_p$, there exists $x_1, x_2 \in \mathcal{X}$, such that $\left|\hat{Y}_{\mathscr{F}}(x_1) - \hat{Y}_{\mathscr{F}^{(1)}}(x_2)\right| > M_p \|x_1 - x_2\|_2$.*

Similar to Theorem 3.1, Theorem 4.1 shows that the expected outcome is not continuous at $x_0$ satisfying $l(x_0) = t_0 - \mathcal{C}$. When we use a randomized classifier, of which the distribution of thresholds has no impulse component, the individual fairness with respect to outcome can be satisfied. The following theorem illustrates this point.

**Theorem 4.2.** *Consider a $d$-dimensional binary classification problem, and let $C_l = \max_x \|\nabla_x l(x)\|_2$, and $\mathcal{F} = \{f(x;t)|t \in (C + \mathcal{C}, D)\}$. If $p(t) \leq L_p$ with $L_p = \frac{M_p}{C_l}$, then for constant $M_p$, we have, $\left|\hat{Y}_{\mathscr{F}}(x_1) - \hat{Y}_{\mathscr{F}}(x_2)\right| \leq M_p \|x_1 - x_2\|_2, \forall x_1, x_2$.*

Now we can see that the requirement for individual fairness w.r.t. outcome is very similar to individual fairness w.r.t. BRC. To find the optimal distribution of $p(t)$ for individual fairness w.r.t. outcome, we can solve a similar optimization problem to (13) by replacing $p(t) \leq L_c$ constraint with $p(t) \leq L_p$. Our theorem also provides the possibility of achieving individual fairness w.r.t. BRC and outcome at the same time by using constraint $0 \leq p(t) \leq \min(L_c, L_p)$.

## 5 Extension to Group Fairness

The analysis of our paper demonstrates the potential usage of randomized classifiers when pursuing individual fairness. In real-world applications, people are often concerned not only with individual fairness, but also with fairness across different groups [28]. There are several methods used to achieve group fairness in the literature [20, 29, 30]. However, individual fairness and group fairness can conflict in some situations [31]. This proposes the question that whether an individually fair randomized classifier can achieve group fairness. To answer this question, we extend the binary classification problem in Section 2 to multi-group case. The following definition and assumptions will be used in this section.

**Definition 5.1** (Multi-group $d$-dimensional binary classification problem)**.** *A $d$-dimensional binary classification problem is a problem of finding a classifier $f : \mathcal{X} \times \mathcal{A} \to \mathcal{Y}$ with $\mathcal{X} \subseteq \mathbb{R}^d$, where $\mathcal{A}$ is the domain of sensitive attribute which is often a discrete set. We denote the features from group $a \in \mathcal{A}$ as $X_a$.*

**Assumption 5.1.** *In a multi-group $d$-dimensional binary classification problem, we assume there are functions $l_a(x)$ for each group such that*

$$l_a(x_1) \geq l_a(x_2) \quad \text{iff} \quad \Pr\{Y = 1|X = x_1, A = a\} \geq \Pr\{Y = 1|X = x_2, A = a\}$$

*and*

$$l_a(x_1) \leq l_a(x_2) \quad \text{iff} \quad \Pr\{Y = 1|X = x_1, A = a\} \leq \Pr\{Y = 1|X = x_2, A = a\}$$

*We further assume that $l_a(x)$ are continuous differentiable functions and $l(X_a) \in [C_a, D_a]$.*

For multi-group cases, we assume the cost function $g(\cdot)$ is different for each groups, which are denoted as $g_a(\cdot)$. For simplicity, we denote the constants $(g_a)^{-1}(\frac{1}{\lambda})$ as $\mathcal{C}_a$. An important group

fairness metric is statistical parity, which requires that the classifier has the same positive outcome rate over all groups, i.e.,

$$\Pr\{\mathscr{F}(X) = 1 | A = a\} = \Pr\{\mathscr{F}(X) = 1 | A = a'\}, \quad \forall a, a' \in \mathcal{A}.$$

In this part, our goal is to find one classifier for each group to ensure statistical parity. We use $p_a(t)$ to denote the distribution of threshold for group $A = a$. Denote the probability density function of $l_a(x)$ as $p_L^a(l)$. Given $p_a(t)$, the overall distribution of $\Pr\{\mathscr{F}(X) = 1 | A = a\}$ is given by,

$$\Pr\{\mathscr{F}(X) = 1 | A = a\} = \int_{C_a + \mathcal{C}_a}^{D_a} \left( \int_{t-\mathcal{C}}^{D_a} p_L^a(l) \mathrm{d}l \right) p_a(t) \mathrm{d}t. \tag{18}$$

Denote the group-specific error rate as $e_a(t)$ which is calculated as follows,

$$e_a(t) = \int_{C_a}^{t-\mathcal{C}_a} \Pr\{Y = 1 | l_a(X) = l\} p_L^a(l) \mathrm{d}l + \int_{t-\mathcal{C}_a}^{D_a} \Pr\{Y = 0 | l_a(X) = l\} p_L^a(l) \mathrm{d}l, \tag{19}$$

we can find optimal $p_a(t)$ by solving the optimization problem

$$p_a(t) = \arg\min_{p_a(t)} \sum_{a \in \mathcal{A}} \left[ \int_{C_a + \mathcal{C}_a}^{D_a} e_a(t) p_a(t) \mathrm{d}t \right] \cdot \Pr\{A = a\},$$

$$s.t. \int_{C + \mathcal{C}_a}^{D_a} p_a(t) = 1, \quad 0 \leq p_a(t) \leq L, \quad \forall a \in \mathcal{A},$$

$$\int_{C_a + \mathcal{C}_a}^{D_a} \left( \int_{t-\mathcal{C}_a}^{D_a} p_L^a(l) \mathrm{d}l \right) p_a(t) \mathrm{d}t = \int_{C_{a'} + \mathcal{C}_{a'}}^{D_{a'}} \left( \int_{t-\mathcal{C}_{a'}}^{D_{a'}} p_L^{a'}(l) \mathrm{d}l \right) p_{a'}(t) \mathrm{d}t, \quad \forall a, a' \in \mathcal{A}. \tag{20}$$

where $L$ could be $L_c$ or $L_p$ from Theorem 3.2 or Theorem 4.2. The objective function is the weighted average of error rate across different groups. To solve this problem, we consider an approximated solution where all $p_a(t)$ are piecewise constant functions. Given a hyper-parameter $K$, we assume that $(C_a + \mathcal{C}_a, D_a)$ can be divided into $K$ bins with the $k$-th bin corresponding to $(C_a + \mathcal{C}_a + \frac{(k-1)(D_a - C_a - \mathcal{C}_a)}{K}, C_a + \mathcal{C}_a + \frac{k(D_a - C_a - \mathcal{C}_a)}{K}]$ and $p_a(t)$ has a constant value in each bin. We denote these constant values as $\{p_{a1}, p_{a2}, ..., p_{aK}\}$, and for simplicity, we denote the start point of the $k$-th bin as $s_{ak}$, i.e. $s_{ak} = C_a + \mathcal{C}_a + \frac{(k-1)(D_a - C_a - \mathcal{C}_a)}{K}$. The last constraint in (20) can be re-written as follows,

$$\sum_{k=1}^{K} p_{ak} \int_{s_{ak}}^{s_{a(k+1)}} \left( \int_{t-\mathcal{C}_a}^{D_a} p_a(l) \mathrm{d}l \right) \mathrm{d}t = \sum_{k=1}^{K} p_{a'k} \int_{s_{a'k}}^{s_{a'(k+1)}} \left( \int_{t-\mathcal{C}_{a'}}^{D_{a'}} p_{a'}(l) \mathrm{d}l \right) \mathrm{d}t, \quad \forall a, a' \in \mathcal{A}. \tag{21}$$

Define $B_{ak} = \int_{s_{ak}}^{s_{a(k+1)}} \left( \int_{t-\mathcal{C}_a}^{D_a} p_L^a(l) \mathrm{d}l \right)$, and similarly define $A_{ak} = \int_{s_{ak}}^{s_{a(k+1)}} \left( \int_{C_a}^{t-\mathcal{C}_a} \rho_1^a(l) \mathrm{d}l + \int_{t-\mathcal{C}_a}^{D_a} \rho_0^a(l) \mathrm{d}l \right) \mathrm{d}t$, where $\rho_1^a(l) = \Pr\{Y = 0 | l_a(X) = l\} p_L^a(l)$, $\rho_0^a(l) = \Pr\{Y = 0 | l_a(X) = l\} p_L^a(l)$. Then problem (20) can be re-written as,

$$\min_{a \in \mathcal{A}, \ 0 \leq k \leq K} \sum_{a \in \mathcal{A}} \sum_{k=1}^{K} A_{ak} p_{ak} \Pr\{A = a\}, \quad s.t. \sum_{k=1}^{K} \frac{(D_a - C_a - \mathcal{C}_a) p_{ak}}{K} = 1, \ 0 \leq p_{ak} \leq L, \forall a \in \mathcal{A},$$

$$\sum_{k=1}^{K} B_{ak} p_{ak} = \sum_{k=1}^{K} B_{a'k} p_{a'k}, \quad \forall a, a' \in \mathcal{A} \tag{22}$$

In practice, the statistical parity constraint can be too strong such that the optimization problem has no solution. Therefore, it can be relaxed as $\left| \sum_{k=1}^{K} B_{ak} p_{ak} - \sum_{k=1}^{K} B_{a'k} p_{a'k} \right| \leq \Omega, \forall a, a' \in \mathcal{A}$. The value of $\Omega$ can be used to control the extent of statistical parity.

Apart from statistical parity, equal opportunity and equalized odds are also commonly used group fairness metrics. For prediction, equal opportunity [15] requires that

$$\Pr\{\mathscr{F}(X) = 1 | Y = 1, A = a\} = \Pr\{\mathscr{F}(X) = 1 | Y = 1, A = a\} \quad \forall a, a' \in \mathcal{A}. \tag{23}$$

Equalized Odds [32] implies that

$$\Pr\{\mathscr{F}(X) = 1 | Y = y, A = a\} = \Pr\{\mathscr{F}(X) = 1 | Y = y, A = a'\} \quad \forall a, a' \in \mathcal{A}, y \in \mathcal{Y}. \tag{24}$$

The optimization problem (22) can be slightly modified to satisfy equal opportunity or equalized odds.

## 6 Experiment

**Dataset and Implementation.** In this section, we conduct experiments on two real-world datasets to evaluate the effectiveness of our proposed methods. The first dataset is the FICO dataset, pre-processed by [15]. This dataset provides the cumulative distribution of credit scores across different racial groups. In our experiments, we generate 5,000 data points according to the distributions for the racial groups *Black* and *Non-Hispanic White*. Each feature vector is defined as $x := [\kappa, a]$, where $\kappa$ denotes the credit score and $a$ represents the individual's race. We assume $l(x) = \kappa$, and define the individual utility function as $\mathscr{U}_{indiv}(f) = [f(x') - f(x)] - 100(l(x') - l(x))^2$, with the associated cost function given by $c(x, x') = 100(l(x') - l(x))^2$.

The second dataset is the Law School Dataset [33], which contains 18,692 student records. We use the same pre-processed version employed in the experiments of [16]. The prediction task is to determine whether a student will pass the bar exam. In our experiment, we include all features excluding *zgpa* and the target variable *bar_pass* in the feature vector $x$. The *zgpa* variable, representing a student's final law school GPA, is used as $l(x)$.[3] To estimate $l(x)$ in practice, we train a linear regression model using the remaining features $x$ to predict zgpa. When evaluating group fairness, we consider race as the sensitive attribute. We define the individual utility function as $\mathscr{U}_{indiv}(f) = [f(x') - f(x)] - (l(x') - l(x))^2$, with the corresponding cost function $c(x, x') = (l(x') - l(x))^2$.

We split the dataset into train/validation/test datasets randomly with a ratio of $60\%/20/\%20$. The baseline model for individual fairness experiment (Tables 1 and 2) is the best deterministic classifier. For the group fairness experiments (Tables 3 and 4), the baseline model consists of two deterministic thresholds (one for each group). We find the deterministic thresholds using grid search on all possible combinations which will maximize the macro average F1 score while satisfying group fairness constraints. For the Law School dataset, we set the number of bins $K$ to 80 and set it to 200 for FICO dataset. Apart from using F1 score to evaluate the accuracy, we use IF ratio for evaluating individual fairness, Statistical Disparity (S-DP), Equal Opportunity Disparity (EO-DP) and Equalized Odds Disparity (ED-DP) for evaluating group fairness. Assume the dataset is $\{x_i, y_i\}_{i=1}^N$ (sensitive attribute $a_i$ is included in $x_i$), IF ratio is defined as $\max_{i \neq j, i, j \in \{1,...,N\}} \frac{|\gamma_i - \gamma_j|}{\|x_i - x_j\|_2}$. For the deterministic classifier, $\gamma_i$ is the predicted outcome or the BRC for data $x_i$. For the randomized classifier, $\gamma_i$ is the expected outcome or expected BRC. S-DP is $|\Pr\{\hat{y}_i = 1 | a_i = 1\} - \Pr\{\hat{y}_i = 1 | a_i = 0\}|$. EO-DP is defined as $|\Pr\{\hat{y}_i = 1 | a_i = 1, y_i = 1\} - \Pr\{\hat{y}_i = 1 | a_i = 0, y_i = 1\}|$, where the probability is estimated as the ratio in the dataset, and ED-DP is the maximum conditional disparity across both positive and negative classes $\max\{|\Pr\{\hat{y}_i = 1 | a_i = 1, y_i = 1\} - \Pr\{\hat{y}_i = 1 | a_i = 0, y_i = 1\}|, |\Pr\{\hat{y}_i = 1 | a_i = 1, y_i = 0\} - \Pr\{\hat{y}_i = 1 | a_i = 0, y_i = 0\}|\}$.

**Results.** Table 1 and Table 2 show the results when using randomized classifiers for the two datasets with different upper bounds on the distributions. IF ratio is measured w.r.t. the prediction [4]. For the

Table 1: Results on FICO dataset applying our randomized classifier

| Method | Deterministic Classifier | Randomized Classifier | | | |
| --- | --- | --- | --- | --- | --- |
| | | $L_p = 1$ | $L_p = 0.5$ | $L_p = 0.25$ | $L_p = 0.1$ |
| F1 score | $0.986 \pm 0.001$ | $0.986 \pm 0.001$ | $0.986 \pm 0.001$ | $0.984 \pm 0.002$ | $0.975 \pm 0.003$ |
| IF ratio | $4.421 \pm 0.691$ | $1.000 \pm 0.000$ | $0.500 \pm 0.000$ | $0.250 \pm 0.000$ | $0.100 \pm 0.000$ |
| S-DP | $0.355 \pm 0.020$ | $0.356 \pm 0.019$ | $0.356 \pm 0.018$ | $0.352 \pm 0.017$ | $0.355 \pm 0.015$ |
| EO-DP | $0.016 \pm 0.011$ | $0.016 \pm 0.010$ | $0.019 \pm 0.010$ | $0.016 \pm 0.006$ | $0.026 \pm 0.011$ |
| ED-DP | $0.016 \pm 0.011$ | $0.016 \pm 0.010$ | $0.019 \pm 0.010$ | $0.016 \pm 0.006$ | $0.026 \pm 0.011$ |

FICO dataset, we can improve individual fairness for a large extent (4.421 to 0.100) with only a small accuracy drop. The IF ratio is equals to the upper bounds $L_p$, which is consistent to Theorem 4.2. For the Law School dataset, we observe that with larger $L_p$, we have a smaller IF ratio with smaller F1 score. IF ratio is not equal to $L_p$ because of the gradient introduced by the linear regression model as discussed in Theorem 4.2. We also measure the group fairness metrics, which shows that there is no direct relationship between individual fairness and group fairness. Therefore, it is possible that we improve individual fairness with a randomized classifier while not exacerbating group unfairness.

---

[3]Figure 1 in Appendix B.1 shows that the estimated $\Pr\{bar\_pass = 1 | zgpa\}$ satisfies Assumption 2.1. A brief description of each attribute is provided in Appendix B.1.

[4]In Appendix C, we displays the results for BRC.

Table 2: Results on Law School dataset applying our randomized classifier

| Method | Deterministic Classifier | Randomized Classifier | | | |
|---|---|---|---|---|---|
| | | $L_p = 1$ | $L_p = 0.8$ | $L_p = 0.4$ | $L_p = 0.3$ |
| F1 score | $0.680 \pm 0.013$ | $0.587 \pm 0.010$ | $0.585 \pm 0.012$ | $0.545 \pm 0.011$ | $0.545 \pm 0.011$ |
| IF ratio | $3.164 \pm 1.671$ | $0.561 \pm 0.008$ | $0.449 \pm 0.007$ | $0.224 \pm 0.003$ | $0.175 \pm 0.005$ |
| S-DP | $0.443 \pm 0.012$ | $0.476 \pm 0.019$ | $0.470 \pm 0.015$ | $0.399 \pm 0.019$ | $0.350 \pm 0.014$ |
| EO-DP | $0.350 \pm 0.028$ | $0.433 \pm 0.041$ | $0.431 \pm 0.034$ | $0.357 \pm 0.040$ | $0.309 \pm 0.041$ |
| ED-DP disparity | $0.362 \pm 0.021$ | $0.433 \pm 0.041$ | $0.431 \pm 0.034$ | $0.357 \pm 0.040$ | $0.309 \pm 0.041$ |

While fixing $L_p = 1$, we validate the effectiveness of our statistical parity constraint in Section 5 by setting $\Omega$ as different values. From Table 3 and 4, the randomized classifiers can have better statistical parity as we decrease $\Omega$. At the same time, we get a lower IF ratio compared to the baseline, which means we improve the group fairness and individual fairness simultaneously. Again, the results show no clear correlation between individual fairness and statistical parity.

Table 3: Results on FICO dataset applying our randomized classifier with statistical parity constraint.

| Method | Deterministic Classifier | Randomized Classifier | | | |
|---|---|---|---|---|---|
| | | $\Omega = 0.1$ | $\Omega = 0.08$ | $\Omega = 0.06$ | $\Omega = 0.04$ |
| F1 score | $0.857 \pm 0.006$ | $0.818 \pm 0.006$ | $0.800 \pm 0.008$ | $0.779 \pm 0.008$ | $0.756 \pm 0.009$ |
| IF ratio | $21.58 \pm 25.87$ | $1.000 \pm 0.000$ | $1.000 \pm 0.000$ | $1.000 \pm 0.000$ | $1.000 \pm 0.000$ |
| S-DP | $0.115 \pm 0.022$ | $0.114 \pm 0.018$ | $0.094 \pm 0.022$ | $0.072 \pm 0.019$ | $0.050 \pm 0.015$ |

Table 4: Results on Law School dataset applying our randomized classifier with statistical parity constraint.

| Method | Deterministic Classifier | Randomized Classifier | | | |
|---|---|---|---|---|---|
| | | $\Omega = 0.1$ | $\Omega = 0.08$ | $\Omega = 0.06$ | $\Omega = 0.04$ |
| F1 score | $0.642 \pm 0.016$ | $0.554 \pm 0.016$ | $0.562 \pm 0.020$ | $0.567 \pm 0.017$ | $0.575 \pm 0.013$ |
| IF ratio | $12.15 \pm 18.11$ | $0.655 \pm 0.065$ | $0.673 \pm 0.056$ | $0.681 \pm 0.055$ | $0.689 \pm 0.062$ |
| S-DP | $0.047 \pm 0.038$ | $0.039 \pm 0.018$ | $0.036 \pm 0.017$ | $0.031 \pm 0.016$ | $0.026 \pm 0.014$ |

## 7 Conclusion

In this paper, we prove the existence of unfairness in strategic classification when using deterministic threshold-based classifiers. To address this issue, we propose a novel approach that employs randomized thresholds, ensuring fairness through an optimizable threshold distribution. We show that this distribution can be efficiently determined using linear programming. Furthermore, by appropriately constraining the threshold distribution, we can achieve both individual and group fairness simultaneously maintaining computational efficiency through linear programming.

## 8 Limitations

While our randomized classifiers offer significant advantages, it is important to note that the proposed method is applicable only when the classification problem can be effectively addressed using a threshold-based classifier. A key assumption is the existence of a function $l(x)$ such that the cost depends solely on the distance between $l(x_1)$ and $l(x_2)$. If this assumption is violated, the method may no longer be valid. Additionally, when the dataset is too small, estimation errors can negatively affect performance. The time complexity of solving a linear optimization problem with $K$ variables using the simplex algorithm can be exponential, i.e. $\mathcal{O}(2^K)$ [34], which may limit the number of bins that can be used in practice.

## 9 Acknowledgment

This work is supported by the U.S. National Science Foundation under award IIS-2301599, CMMI-2301601, and DMS-2529302 and by grants from the Ohio State University's Translational Data Analytics Institute and College of Engineering Strategic Research Initiative.

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

# A Proofs

## A.1 Proof for Theorem 3.1

*Proof.* When $t_0 \in (C + \mathcal{C}, D)$, we can find $x_1$ such that $l(x_1) = t_0 - \mathcal{C}$ in $\mathcal{X}$. From the definition of BRC, we have $c_f(x_1) = \frac{1}{\lambda}$. For any positive constant $\epsilon$, we know that for $x_2$ satisfying $l(x_2) = t_0 - \mathcal{C} - \epsilon$, $c_f(x_2) = 0$. Because $l(x)$ satisfies

$$|l(x_1) - l(x_2)| \geq L_l \|x_1 - x_2\|_2,\tag{25}$$

we have

$$|c_f(x_1) - c_f(x_2)| = \frac{1}{\lambda},\tag{26}$$

$$\|x_1 - x_2\|_2 \leq \frac{\epsilon}{L_l}.\tag{27}$$

When $\epsilon < \frac{L_l}{M_c\lambda}$, we have $|c_f(x_1) - c_f(x_2)| > M_c \|x_1 - x_2\|_2$. $\qquad\square$

## A.2 Proof for Theorem 3.2

*Proof.* For any input $x$, the individual cost is

$$c_{\mathscr{F}}(x) = \int_{l(x)}^{l(x)+\mathcal{C}} g(t - l(x))p(t)\mathrm{d}t.\tag{28}$$

The gradient of $c_{\mathscr{F}}(x)$ is

$$\nabla_x c_{\mathscr{F}}(x) = \nabla_x l(x)\left[g(\mathcal{C})p\left(l(x) + \mathcal{C}\right) - g(0)p(l(x)) - \int_{l(x)}^{l(x)+\mathcal{C}} g'(t - l(x))p(t)\mathrm{d}t\right].\tag{29}$$

From the assumptions that $g(0) = 0$ and $g(\mathcal{C}) = \frac{1}{\lambda}$, we have

$$\|\nabla_x c_{\mathscr{F}}(x)\|_2 = \|\nabla_x l(x)\|_2 \left|\frac{1}{\lambda}p(l(x) + \mathcal{C}) - \int_{l(x)}^{l(x)+\mathcal{C}} g'(t - l(x))p(t)\mathrm{d}t\right|.\tag{30}$$

Since $g(\cdot)$ is strictly increasing, we have $g'(\cdot) > 0$. Therefore,

$$\left|\frac{1}{\lambda}p(l(x) + \mathcal{C}) - \int_{l(x)}^{l(x)+\mathcal{C}} g'(t - l(x))p(t)\mathrm{d}t\right|$$

$$= \begin{cases} \frac{1}{\lambda}p(l(x) + \mathcal{C}) - \int_{l(x)}^{l(x)+\mathcal{C}} g'(t - l(x))p(t)\mathrm{d}t & \text{if } \frac{1}{\lambda}p(l(x) + \mathcal{C}) \geq \int_{l(x)}^{l(x)+\mathcal{C}} g'(t - l(x))p(t)\mathrm{d}t, \\ \int_{l(x)}^{l(x)+\mathcal{C}} g'(t - l(x))p(t)\mathrm{d}t - \frac{1}{\lambda}p(l(x) + \mathcal{C}) & \text{otherwise.} \end{cases}$$

$$\tag{31}$$

- If $\frac{1}{\lambda}p(l(x) + \mathcal{C}) \geq \int_{l(x)}^{l(x)+\mathcal{C}} g'(t - l(x))p(t)\mathrm{d}t$, then

$$\|\nabla_x c_{\mathscr{F}}(x)\|_2 \leq \|\nabla_x l(x)\|_2 \cdot \frac{1}{\lambda}p(l(x) + \mathcal{C}) \leq \frac{C_l L_c}{\lambda}.\tag{32}$$

  If $L_c = \frac{\lambda M_c}{C_l}$, then $\|\nabla_x c_{\mathscr{F}}(x)\|_2 \leq M_c$.

- If $\frac{1}{\lambda}p(l(x) + \mathcal{C}) < \int_{l(x)}^{l(x)+\mathcal{C}} g'(t - l(x))p(t)\mathrm{d}t$, then

$$\|\nabla_x c_{\mathscr{F}}(x)\|_2 \leq \|\nabla_x l(x)\|_2 \int_{l(x)}^{l(x)+\mathcal{C}} g'(t - l(x))p(t)\mathrm{d}t \leq C_l C_g L_c \mathcal{C}.\tag{33}$$

  If $L_c = \frac{M_c}{C_l C_g \mathcal{C}}$, then $\|\nabla_x c_{\mathscr{F}}(x)\|_2 \leq M_c$.

Therefore, $L_c = \min\left\{\frac{\lambda M_c}{C_l}, \frac{M_c}{C_l C_g \mathcal{C}}\right\}$. $\qquad\square$

## A.3 Proof for Theorem 4.1

*Proof.* When $t_0 \in (C + \mathcal{C}, D)$, we can find $x_1 \in \mathcal{X}$ such that $l(x_1) = t_0 - C$. From the definition of $\hat{Y}_{\mathscr{F}}(\cdot)$, we know that

$$\hat{Y}_{\mathscr{F}}(x_1) = f(x_1; t_0) = 1. \tag{34}$$

For any positive constant $\epsilon$, we can find $x_2 \in \mathcal{X}$ satisfying $l(x_2) = t_0 - C - \epsilon$, and we have

$$\hat{Y}_{\mathscr{F}}(x_2) = f(x_2; t_0) = 0. \tag{35}$$

Since $l(x)$ satisfies

$$|l(x_1) - l(x_2)| \geq L_l \|x_1 - x_2\|_2, \tag{36}$$

we obtain

$$\|x_1 - x_2\|_2 \leq \frac{\epsilon}{L_l}. \tag{37}$$

When $\epsilon < \frac{L_l}{M_p}$, we have

$$\left| \hat{Y}_{\mathscr{F}}(x_1) - \hat{Y}_{\mathscr{F}}(x_2) \right| > M_p \|x_1 - x_2\|_2. \tag{38}$$

$\square$

## A.4 Proof for Theorem 4.2

*Proof.* Consider an arbitrary $x_1 \in \mathcal{X}$, we know that $f(x_1; t) = 1$ if and only if there exists $x_1'$, such that

$$l(x_1') \geq t \quad \text{and} \quad l(x_1') - l(x_1) \leq \mathcal{C}. \tag{39}$$

Therefore,

$$\hat{Y}_{\mathscr{F}}(x_1) = \int_{C+\mathcal{C}}^{l(x_1)+\mathcal{C}} p(t)\mathrm{d}t. \tag{40}$$

Again, we have

$$\left| \hat{Y}_{\mathscr{F}}(x_1) - \hat{Y}_{\mathscr{F}}(x_2) \right| = \left| \int_{l(x_1)+\mathcal{C}}^{l(x_2)+\mathcal{C}} p(t)\mathrm{d}t \right|$$
$$\leq L_p |l(x_1) - l(x_2)|. \tag{41}$$

Because $\|\nabla_x l(x)\|_2$ is upper bounded by $C_l$, and $L_p = \frac{M_p}{C_l}$, we have

$$\left| \hat{Y}_{\mathscr{F}}(x_1) - \hat{Y}_{\mathscr{F}}(x_2) \right| \leq M_p \|x_1 - x_2\|_2. \tag{42}$$

$\square$

# B Implementation Details

## B.1 Dataset

The Law School dataset contains 12 attributes. The information about them are displayed in Table 5. Figure 1 shows the estimation of $\Pr\{\text{bar\_pass} = 1 | \text{zgpa}\}$ from the whole dataset. Since the conditional probability is an increasing function w.r.t zgpa, Assumption 2.1 is satisfied. We can regard zgap as $l(x)$.

## B.2 Computational Resources

All experiments are conducted on a server equipped with 64 AMD EPYC 7313 16-Core CPUs. The server also includes 8 NVIDIA RTX A5000 GPUs (24GB each), although GPU resources are not utilized for our experiments.

Each experiment runs efficiently, completing 5 random seeds in under an hour. The optimization of the randomized classifier is computationally inexpensive. However, evaluating the Individual Fairness (IF) ratio is more time-consuming, as it requires comparing all pairs $(x_i, x_j)$ in the dataset $\{(x_i, y_i)\}_{i=1}^N$, resulting in a computational complexity of $\mathcal{O}(N^2)$.

Table 5: The meaning of attributes in the Law School dataset [35].

| Attribute | Description | Data Information |
|-----------|-------------|------------------|
| decile1b | First-year law school GPA decile | integer, ranging from 1 to 10 |
| decile3 | Third-year law school GPA decile | integer, ranging from 1 to 10 |
| lsat | LSAT score | integer, ranging from 0 to 60 |
| ugpa | Undergraduate GPA on a 4.0 scale | real values in [0, 4.0] |
| zfygpa | Standardized first-year GPA (z-score) | real values in [-4.0, 4.0] |
| zgap | Standardized final GPA (z-score) | real values in [-4.0, 4.0] |
| fulltime | Enrollment status | 1: full-times, 0: part-time |
| fam_inc | Family income bracket | integer, ranging from 1.0 to 5.0 |
| male | Gender indicator | 1: male, 0: female |
| racetxt | Race category | binary, not clear the meaning of 0 and 1 |
| tier | Law school tier | integer, ranging from 1 to 6 |
| pass_bar | Bar exam pass indicator | 1: pass, 0: not-pass |

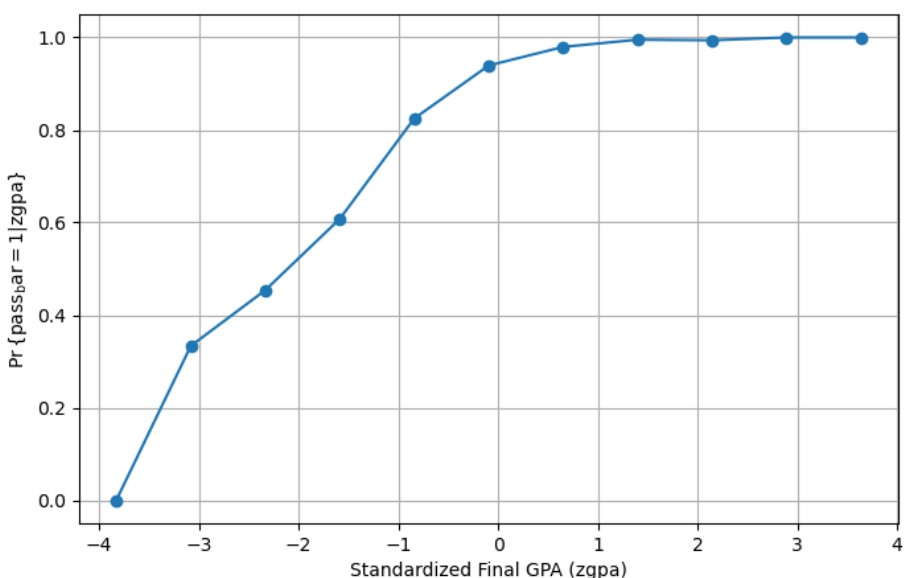

Figure 1: Estimated $\Pr\{bar\_pass = 1|zgpa\}$

## C  Additional Results

In this section, we displays the additional experiment results about measuring individual fairness metric (IF ratio) w.r.t. BRC.

Table 6: Results on FICO dataset applying our randomized classifier with IF ratio measured w.r.t. cost.

| Method | Deterministic Classifier | Randomized Classifier | | | |
|---|---|---|---|---|---|
| | | $L_c = 1$ | $L_c = 0.5$ | $L_c = 0.25$ | $L_c = 0.1$ |
| F1 score | $0.986 \pm 0.001$ | $0.986 \pm 0.001$ | $0.986 \pm 0.001$ | $0.984 \pm 0.002$ | $0.975 \pm 0.003$ |
| IF ratio | $3.412 \pm 1.156$ | $0.866 \pm 0.041$ | $0.450 \pm 0.034$ | $0.227 \pm 0.018$ | $0.090 \pm 0.006$ |
| S-DP | $0.355 \pm 0.020$ | $0.356 \pm 0.019$ | $0.356 \pm 0.018$ | $0.352 \pm 0.017$ | $0.355 \pm 0.015$ |
| EO-DP | $0.016 \pm 0.011$ | $0.016 \pm 0.010$ | $0.019 \pm 0.010$ | $0.016 \pm 0.006$ | $0.026 \pm 0.011$ |
| ED-DP | $0.016 \pm 0.011$ | $0.016 \pm 0.010$ | $0.019 \pm 0.010$ | $0.016 \pm 0.006$ | $0.026 \pm 0.011$ |

Table 7: Results on Law School dataset applying our randomized classifier with IF ratio measured w.r.t. BRC.

| Method | Deterministic Classifier | Randomized Classifier | | | |
|---|---|---|---|---|---|
| | | $L_c = 1$ | $L_c = 0.8$ | $L_c = 0.4$ | $L_c = 0.3$ |
| F1 score | $0.680 \pm 0.013$ | $0.587 \pm 0.010$ | $0.585 \pm 0.012$ | $0.545 \pm 0.011$ | $0.545 \pm 0.011$ |
| IF ratio | $3.776 \pm 3.039$ | $0.270 \pm 0.107$ | $0.150 \pm 0.085$ | $0.057 \pm 0.018$ | $0.036 \pm 0.018$ |
| S-DP | $0.443 \pm 0.012$ | $0.476 \pm 0.019$ | $0.470 \pm 0.015$ | $0.399 \pm 0.019$ | $0.350 \pm 0.014$ |
| EO-DP | $0.350 \pm 0.028$ | $0.433 \pm 0.041$ | $0.431 \pm 0.034$ | $0.357 \pm 0.040$ | $0.309 \pm 0.041$ |
| ED-DP disparity | $0.362 \pm 0.021$ | $0.433 \pm 0.041$ | $0.431 \pm 0.034$ | $0.357 \pm 0.040$ | $0.309 \pm 0.041$ |

Table 8: Results on FICO dataset applying our randomized classifier with statistical parity constraint and IF ratio measured w.r.t. BRC.

| Method | Deterministic Classifier | Randomized Classifier | | | |
|---|---|---|---|---|---|
| | | $\Omega = 0.1$ | $\Omega = 0.08$ | $\Omega = 0.06$ | $\Omega = 0.04$ |
| F1 score | $0.857 \pm 0.006$ | $0.818 \pm 0.006$ | $0.800 \pm 0.008$ | $0.779 \pm 0.008$ | $0.756 \pm 0.009$ |
| IF ratio | $7.105 \pm 5.233$ | $0.868 \pm 0.047$ | $0.853 \pm 0.037$ | $0.829 \pm 0.025$ | $0.796 \pm 0.079$ |
| S-DP | $0.115 \pm 0.022$ | $0.114 \pm 0.018$ | $0.094 \pm 0.022$ | $0.072 \pm 0.019$ | $0.050 \pm 0.015$ |

Table 9: Results on Law School dataset applying our randomized classifier with statistical parity constraint and IF ratio measured w.r.t. BRC.

| Method | Deterministic Classifier | Randomized Classifier | | | |
|---|---|---|---|---|---|
| | | $\Omega = 0.1$ | $\Omega = 0.08$ | $\Omega = 0.06$ | $\Omega = 0.04$ |
| F1 score | $0.642 \pm 0.016$ | $0.554 \pm 0.016$ | $0.562 \pm 0.020$ | $0.567 \pm 0.017$ | $0.575 \pm 0.013$ |
| IF ratio | $12.15 \pm 18.11$ | $0.166 \pm 0.055$ | $0.181 \pm 0.087$ | $0.153 \pm 0.080$ | $0.147 \pm 0.025$ |
| S-DP | $0.047 \pm 0.038$ | $0.039 \pm 0.018$ | $0.036 \pm 0.017$ | $0.031 \pm 0.016$ | $0.026 \pm 0.014$ |

