# OpenReview forum: "Individual Fairness In Strategic Classification"
_NeurIPS.cc/2025/Conference — NeurIPS 2025 poster_

### Official Review · Reviewer_wL1p · 2025-06-20

**Clarity:** 2
**Significance:** 2
**Originality:** 3
**Rating:** 4
**Confidence:** 3

**Summary:**

The authors consider the problem of enforcing individual fairness when deploying threshold classifiers in situations where individuals engage in strategic behavior. They find that deterministic classifiers will fail to enforce individual fairness with respect to both outcome and cost, the authors also show that deploying a randomized classifier can fix this problem and provide algorithms that find an optimal but fair distribution over the thresholds by breaking the distribution $p(t)$ into piecewise constant functions. Results are extended to group fairness and experiments are provided on FICO and law school data sets.

**Questions:**

* Is it unknown if the constraint (20) is feasible to enforce? Can the authors add more discussion on the use of their relatively strong assumptions?

**Ethical Concerns:**

["NO or VERY MINOR ethics concerns only"]

**Final Justification:**

The authors addressed my points in rebuttal

**Limitations:**

yes

**Quality:**

3

**Strengths And Weaknesses:**

**Strengths**
* The authors analysis includes a novel cost-based definition of individual fairness for strategic classification which requires $|C(x_1, f)-C(x_2,f)|<|x_1-x_2|$. This contrasts with the usual outcome based definitions of individual fairness.
* The theory and methodology are evaluated in two real world settings (admissions and credit scoring) where individuals are likely to engage in strategic behavior.

**Weaknesses**

* The theory requires the following assumptions: (i) The existence of a function $l$ s.t. $Pr(Y=1|x) \geq Pr(Y=1|z)$ iff $l(x) \geq l(z)$. (ii) c(x,x') = g(l(x)-l(x')) for some increasing function $g$. (iii) The learner deploys classifier $f(x;t) = 1 [l(\Delta(x(f)) ≥ t]$. (i) is reasonable for many applications but (iii) implicitly requires that the learner know $l$ which is a very strong assumption and (ii) requires that the cost be directly related to l(), but often the features that individuals can game are not the direct cause of the outcome of interest. For example, someones credit often does not directly cause a default in debt down the line.
* Unless I missed it, the group fairness results seem to miss a theorem on the feasibility of the constraint (20).
* The paper is very dense with derivations, including many inline definitions of constants and algebra steps making readability limited.

---

> ### Author Rebuttal · Authors · 2025-07-31
>
> **Weakness 1**: Assumption ii can be relaxed with the following theorem.
>
> **Theorem** : Consider a $d$-dimensional binary classification problem, suppose the cost function $c(x, x')$ satisfies  $0 \leq c(x, x') < \infty$, $c(x, x) = 0$ and  Lipschitz continuouity w.r.t. $x$, i.e.
>
> $$|c(x_{1}, x') - c(x_{2}, x')| \leq C_{c} ||x_{1} - x_{2}||,  \forall x_{1}, x_{2},$$
>
> and $\mathcal{F} = \\{f(x; t)|t \in [C, D]\\}$ for randomized classifier $\mathscr{F}$. Denote the best response as $r(x, t)$ and
>
> $$ r(x;t) = \begin{cases}
>             \arg\min_{x'}\\{c(x,x') | l(x') \geq t\; c(x,x') \le \tfrac{1}{\lambda}\\},
>             & \text{if a solution exists}, \\\\
>             x, & \text{otherwise}.
> \end{cases}.
> $$
>
> Assume $\arg\min_{x'}\\{c(x,x') | l(x') \geq t, c(x,x') \leq \tfrac{1}{\lambda}\\}$ has a unique solution when the solution exists, $r(x, t)$ is bounded and the set of points where $r(x, t)$ is discontinuous has measure zero. If $p(t) \leq L_{c}$ with $L_{c} = \frac{M_{c}}{C_{c}(D - C)}$, then for constant $M_{c}$, $\forall x_{1}, x_{2} \in \mathcal{X}$, we have,
>     $$
>         |c_{\mathscr{F}}(x_{1}) - c_{\mathscr{F}}(x_{2})| \leq M_{c}||x_{1} - x_{2}||.
>     $$
>
>
> For assumption (iii), we can use the approximated function of $l(x)$ in the classification. And learning the score function is the central task for many applications.
>
>
>
> **Weakness 2 and Q1**: Constraint 20 is not always feasible to enforce. We also mentioned this problem in Line 228 of the problem. In this case, we can relax the constraints to achieve a trade-off between accuracy, individual fairness and statistical parity as we stated in the paper.
>
>
>
> **Weakness 3** : We will reduce the inline definitions and algebra steps in the next version of the paper.

---

> > ### Comment · Reviewer_wL1p · 2025-08-01
> >
> > Thank you for this discussion. I maintain my score to accept.

---

> > > ### Author Response · Authors · 2025-08-01
> > >
> > > Thank you so much for your valuable comments and your engagement in discussion.

---

### Official Review · Reviewer_xwZK · 2025-07-01

**Clarity:** 4
**Significance:** 3
**Originality:** 4
**Rating:** 4
**Confidence:** 2

**Summary:**

This paper studies individual fairness in strategic classification, where individuals may modify their features to influence machine learning decisions. The authors demonstrate that deterministic threshold-based classifiers inherently violate individual fairness in such settings, particularly concerning the cost individuals incur to adjust their features. To address this, they propose randomized threshold classifiers, proving under certain distribution conditions that these classifiers can guarantee individual fairness while maintaining high accuracy. They design these classifiers as an efficient linear programming problem and extend the method to incorporate group fairness notions. Experiments on the FICO and Law School datasets show that the proposed method significantly improves individual fairness and group fairness simultaneously.

**Questions:**

- Given the method relies on threshold-based classifiers and specific assumptions on l(x), could the authors give an example of scenarios where these assumptions might not hold and discuss whether the method could be adapted to such cases?

- How sensitive is the method’s performance (fairness vs. accuracy) to hyperparameters like the number of bins k. This could help to tune the method effectively.

- Could the authors elaborate on the practical structure or interpretability of the learned threshold distribution p(t)? For example, does p(t) typically concentrate around certain thresholds, and how should this be interpreted in terms of fairness-accuracy trade-offs?

Very nice work !

**Ethical Concerns:**

["NO or VERY MINOR ethics concerns only"]

**Final Justification:**

The authors have addressed most of my concerns with detailed theoretical clarifications and additional empirical results.
While the approach is still limited to threshold-based classifiers and evaluated on two datasets, I believe the paper makes a meaningful and novel contribution to the underexplored area of individual fairness under strategic behavior.

**Limitations:**

Yes. The authors explicitly discuss limitations regarding reliance on threshold-based classifiers, assumptions on the existence of a score function l(x), small datasets, and simplex solver complexity.

**Paper Formatting Concerns:**

No. All good.

**Quality:**

4

**Strengths And Weaknesses:**

Strengths:
- This study addresses the underexplored area of individual fairness.
- It proves when individual fairness is impossible with deterministic thresholds and how to achieve it using randomized thresholds.
- It presents a linear programming formulation for designing fair classifiers and extends the approach to incorporate group fairness simultaneously with individual fairness.
- The approach is tested on two datasets with a detailed quantitative analysis.

Weaknesses:
- The approach relies on threshold-based classifiers and assumptions on l(x), which limits its applicability to broader ML settings.
- The experiments are limited to 2 datasets and do not test sensitivity to hyperparameters such as the number of bins (k), which could generate computational challenges if larger values are required.

---

> ### Author Rebuttal · Authors · 2025-07-31
>
> **Weakness 1 and Q1**: The assumptions on $l(x)$ (Eqs. 1 and 2) ensure that decisions can be made based on a score function. In theory, any classification problem can be solved by a threshold-based classifier when $l(x)$ is allowed to be an arbitrary function. However, in practice, such a function may not be learnable. The connection between cost function and $l(x)$ can be relaxed with the following theorem.
>
>
> **Theorem** : Consider a $d$-dimensional binary classification problem, suppose the cost function $c(x, x')$ satisfies  $0 \leq c(x, x') < \infty$, $c(x, x) = 0$ and  Lipschitz continuouity w.r.t. $x$, i.e.
>
> $$|c(x_{1}, x') - c(x_{2}, x')| \leq C_{c} ||x_{1} - x_{2}||,  \forall x_{1}, x_{2},$$
>
> and $\mathcal{F} = \\{f(x; t)|t \in [C, D]\\}$ for randomized classifier $\mathscr{F}$. Denote the best response as $r(x, t)$ and
>
> $$ r(x;t) = \begin{cases}
>             \arg\min_{x'}\\{c(x,x') | l(x') \geq t\; c(x,x') \le \tfrac{1}{\lambda}\\},
>             & \text{if a solution exists}, \\\\
>             x, & \text{otherwise}.
> \end{cases}.
> $$
>
> Assume $\arg\min_{x'}\\{c(x,x') | l(x') \geq t, c(x,x') \leq \tfrac{1}{\lambda}\\}$ has a unique solution when the solution exists, $r(x, t)$ is bounded and the set of points where $r(x, t)$ is discontinuous has measure zero. If $p(t) \leq L_{c}$ with $L_{c} = \frac{M_{c}}{C_{c}(D - C)}$, then for constant $M_{c}$, $\forall x_{1}, x_{2} \in \mathcal{X}$, we have,
>     $$
>         |c_{\mathscr{F}}(x_{1}) - c_{\mathscr{F}}(x_{2})| \leq M_{c}||x_{1} - x_{2}||.
>     $$
>
> Our method of randomization can still be used to improve individual fairness. However, how to get the optimal distribution of $t$ depends on the specific problem.
>
> **Weakness 2 and Q2**: Since finding an optimal distribution of $p(t)$ is a linear optimization problem, the worst time complexity can be $\mathcal{O}(2^K) $. However, in practice, the worst case does not happen and algorithms such as the simplex method typically solve problems in polynomial time for most cases.  We include the following experiments to test the sensitivity against the number of bins, which show that the results are not very sensitive to the number of bins in this case.
>
> |Metric |#bins = 50| #bins = 100|#bins = 150|#bins = 200|#bins = 250|
> |---|---|---|---|---|---|
> |F1 score |0.978 $\pm$ 0.002|0.984 $\pm$ 0.003| 0.985 $\pm$ 0.003|0.986 $\pm$ 0.001| 0.985 $\pm$ 0.002|
> |IF ratio |0.408 $\pm$ 0.046 |0.432 $\pm$ 0.040 |0.461 $\pm$ 0.016 |0.450 $\pm$ 0.034 |0.452 $\pm$ 0.014 |
> |S-DP|0.357 $\pm$ 0.016 |0.355 $\pm$ 0.017 |0.355 $\pm$ 0.017 |0.356 $\pm$ 0.018 |0.355 $\pm$ 0.015 |
> |EO-DP |0.032 $\pm$ 0.011 |0.014 $\pm$ 0.008 |0.014 $\pm$ 0.011 |0.019 $\pm$ 0.010 |0.018 $\pm$ 0.011|
> |ED-DP |0.032 $\pm$ 0.011 |0.014 $\pm$ 0.008 |0.014 $\pm$ 0.011 |0.019 $\pm$ 0.010 |0.018 $\pm$ 0.011|
>
> **Q3**: The distribution is usually centered at the optimal deterministic threshold, but the specific shape is affected by the distribution of feature $x$. In terms of accuracy-fairness tradeoff, the best accuracy is achieved when the distribution of a delta function is at the optimal threshold. When the distribution spreads out, we get better fairness, but more misclassifications will happen.

---

> > ### Author Response · Authors · 2025-08-04
> >
> > Dear Reviewer,
> >
> > Thank you for the time and effort you devoted to reviewing our work. We have carefully considered all of your comments and prepared a detailed, point‑by‑point response. We would greatly value it if you could review our replies to see whether they resolve your concerns. We are glad to provide further clarification.

---

> ### Comment · Reviewer_xwZK · 2025-08-05
>
> Dear authors,
> Thank you for your responses and clarifications. Most of my concerns are being addressed.
> I maintain my score to borderline accept.

---

> > ### Author Response · Authors · 2025-08-06
> >
> > We sincerely thank you for the thoughtful evaluation of our work. We are grateful for your constructive feedback. We remain fully available and would be happy to elaborate on any aspect of our work if you need.

---

### Official Review · Reviewer_hq9t · 2025-07-02

**Clarity:** 3
**Significance:** 2
**Originality:** 3
**Rating:** 4
**Confidence:** 3

**Summary:**

This paper considers individually fair decision-making settings in which individuals can adjust their features (by paying some cost). Assuming that each individual will behave optimally, constraints for individual fairness are derived and a randomized classifier with maximal accuracy is constructed using linear programming. Evaluations on Law School and FICO datasets show that individual fairness can be provided by this approach with reasonable decreases in predictive performance.

**Questions:**

Q1. Why are the costs of feature adjustments assumed to be related to the l function in Eq. (7)? Is the method strongly dependent on this assumption (i.e., necessary for individual fairness, or computationally needed)? It seems that in real applications, the cost would simply be domain specific (and given) and generally unrelated to the prediction function. If given an arbitrary cost function, what could be done using your approach?

Q2. It seems individual fairness requires randomized classifiers in general (i.e., in non-strategic settings). Why is there such a strong emphasis on this randomization in the paper?

Q3. In (8), what does the left-hand side notation mean? Should (3) also match this notation?

Q4. Are there smoother probability baselines (e.g., based on logistic regression) to compare against?

**Ethical Concerns:**

["NO or VERY MINOR ethics concerns only"]

**Final Justification:**

The author response addressed my main concerns about the generality/assumptions of the approach.

**Limitations:**

yes

**Quality:**

3

**Strengths And Weaknesses:**

Strengths:
+ This paper identifies an important gap in fair machine learning research: individual fairness in strategic classification settings.
+ The approach is novel, leveraging a distribution over linear threshold functions obtained using a linear program to provide individual fairness.
+ The approach is versatile and can be combined with group fairness criteria, as the authors demonstrate.
+ The experimental results show a good trade-off between predictive performance and providing individual fairness.

Weaknesses:
- The assumptions about the cost function (relating specifically to the linear threshold function) appear unnatural to me (see Q1), decreasing the general applicability of the approach to only work in settings with very strict assumptions.
-  The framing of the paper seems a little strange to me. I believe individual fairness generally requires randomized decisions, so the focus on this aspect in the strategic setting is unnecessary (see Q2).
- Related to the last point, since (I believe) randomization is essential for individual fairness, this suggests that better randomized baselines should be employed (Q3) rather than the deterministic classifier.

----

There is a small typo in the Section 3 header.

---

> ### Author Rebuttal · Authors · 2025-07-31
>
> **Weakness 1 & Q1**:  Equation 7 captures scenarios where the cost of improving user features depends on the actual improvement achieved by the action. Here, $l(x)$ is not a prediction function but a function related to the ground-truth label (see Eqs. 1 and 2), and the relationship between cost and $l(x)$ is also employed in [1].
>
> For a more general cost function, we can have the following theorem.
>
> **Theorem** : Consider a $d$-dimensional binary classification problem, suppose the cost function $c(x, x')$ satisfies  $0 \leq c(x, x') < \infty$, $c(x, x) = 0$ and  Lipschitz continuouity w.r.t. $x$, i.e.
>
> $$|c(x_{1}, x') - c(x_{2}, x')| \leq C_{c} ||x_{1} - x_{2}||,  \forall x_{1}, x_{2},$$
>
> and $\mathcal{F} = \\{f(x; t)|t \in [C, D]\\}$ for randomized classifier $\mathscr{F}$. Denote the best response as $r(x, t)$ and
>
> $$ r(x;t) = \begin{cases}
>             \arg\min_{x'}\\{c(x,x') | l(x') \geq t\; c(x,x') \le \tfrac{1}{\lambda}\\},
>             & \text{if a solution exists}, \\\\
>             x, & \text{otherwise}.
> \end{cases}.
> $$
>
> Assume $\arg\min_{x'}\\{c(x,x') | l(x') \geq t, c(x,x') \leq \tfrac{1}{\lambda}\\}$ has a unique solution when the solution exists, $r(x, t)$ is bounded and the set of points where $r(x, t)$ is discontinuous has measure zero. If $p(t) \leq L_{c}$ with $L_{c} = \frac{M_{c}}{C_{c}(D - C)}$, then for constant $M_{c}$, $\forall x_{1}, x_{2} \in \mathcal{X}$, we have,
>     $$
>         |c_{\mathscr{F}}(x_{1}) - c_{\mathscr{F}}(x_{2})| \leq M_{c}||x_{1} - x_{2}||.
>     $$
>
> **Proof** : The individual cost is
>
> $$
> c_{\mathscr{F}}(x_{1}) = \int_{C}^{D} c(x_{1}, r(x_{1}; t))p(t)\mathrm{d}t.
> $$
>
> Therefore,
>
> $$
> |c_{\mathscr{F}}(x_{1}) - c_{\mathscr{F}}(x_{2})| \leq \int_{C}^{D}|c(x_{1}, r(x_{1}; t)) - c(x_{2}, r(x_{2}; t))|p(t)\mathrm{d}t.
> $$
>
> Because
>
> $$
> c(x_{1}, r(x_{1}; t)) \leq c(x_{1}, r(x_{2}; t)), ~~~ c(x_{2}, r(x_{2}; t)) \leq c(x_{2}, r(x_{1}; t)),
> $$
>
> we have
>
> $$
>  |c(x_{1}, r(x_{1}; t)) - c(x_{2}, r(x_{2}; t))|
> \leq  \max \\{|c(x_{1}, r(x_{2}; t)) - c(x_{2}, r(x_{2}; t))|, |c(x_{2}, r(x_{1}; t)) - c(x_{1}, r(x_{1}; t))|\\}
> \leq  C_{c}||x_{1} - x_{2}||.
> $$
>
> With $p(t) \leq \frac{M_{c}}{C_{c}(D - C)}$,
>
> $$
> |c_{\mathscr{F}}(x_{1}) - c_{\mathscr{F}}(x_{2})| \leq M_{c}||x_{1} - x_{2}||.
> $$
>
> **Weakness 2 & Q2**: We acknowledge that randomization is a general approach to achieving individual fairness. The motivation of this paper is to investigate individual fairness in the context of strategic classification, where agents may adjust their features. Prior research has shown that many fairness methods effective in static settings fail to extend naturally to strategic scenarios [2, 3, 4]. Moreover, cost, which is intrinsic to strategic classification, has been an important topic in the literature. Therefore, it is both timely and valuable to explore how randomization can be applied to strategic classification.
>
> In non-strategic settings, where no cost is involved, individual fairness with respect to outcomes can be viewed as a special case of Section 4, where $\mathcal{C} = 0$ and $\Delta(x) = x, \forall x$.
>
> **Q3**: The left-hand side represents the $l$ function evaluated on the adjusted feature, where the original feature is $x$ and the adjustment is determined by the function $f(x, t)$. Thank you for pointing out the inconsistency. To align with (3) and emphasize the threshold used in $f$, it should be written as $l(\Delta x(f(\cdot; t)))$.
>
> **Weakness 4 & Q4**: As we discussed in the response to weakness 2 & Q2, strategic classification has its own specific property. Smoother baselines, logistic regression, has the problem of how an individual will react to them, which is beyond the scope of this paper.
>
> In our experiments on Law School dataset, we compared our method against a baseline which is very similar to logistic regression where we learn a linear function on $x$ and then find the best thresholds.
>
> In static settings, the randomized logistic regression classifier $\Pr\\{\hat{Y} = 1| X = x\\} = \frac{1}{1 + e^{-w^{T}x + b}}$ can be seen as a special case of our randomized classifier with $l(x) = -w^{T}x + b$ and $\Pr\\{t \leq l\\} = \frac{1}{1 + e^{-l}}$. Therefore, our method can be used to improve individual fairness further based on $l(x)$ learned by logistic regression model. Here we provide a comparison on the Law school dataset in static setting. Our method used 300 bins.
>
> |Metric|F1 score|IF ratio|S-DP| EO-DP| ED-DP|
> |---|---|---|---|---|---|
> |Logistic regression |0.627 $\pm$ 0.007| 0.232 $\pm$ 0.037|0.304 $\pm$ 0.016|0.211 $\pm$ 0.031|0.301 $\pm$ 0.031|
> |Our method $L_{p} = 0.2$|0.635 $\pm$ 0.008| 0.212 $\pm$ 0.030|0.293 $\pm$ 0.046|0.219 $\pm$ 0.065|0.270 $\pm$ 0.033|
> |Our method $L_{p} = 0.15$|0.586 $\pm$ 0.022|0.159 $\pm$ 0.023|0.310 $\pm$ 0.029|0.248 $\pm$ 0.049|0.259 $\pm$ 0.041|
>
> If you have any other smoother baselines that can be used here, we would be very glad to provide a comparison during the discussion.
>
>
>
> [1] Milli, Smitha, et al. "The social cost of strategic classification." Proceedings of the conference on fairness, accountability, and transparency. 2019.
>
> [2] Diana, Emily, Saeed Sharifi-Malvajerdi, and Ali Vakilian. "Minimax group fairness in strategic classification." 2025 IEEE Conference on Secure and Trustworthy Machine Learning (SaTML). IEEE, 2025.
>
> [3] Keswani, Vijay, and L. Elisa Celis. "Addressing strategic manipulation disparities in fair classification." Proceedings of the 3rd ACM Conference on Equity and Access in Algorithms, Mechanisms, and Optimization. 2023.
>
> [4] Estornell, Andrew, et al. "Unfairness despite awareness: group-fair classification with strategic agents." arXiv preprint arXiv:2112.02746 (2021).

---

> > ### Author Response · Authors · 2025-08-04
> >
> > Dear Reviewer,
> >
> > Thank you very much for taking the time to review our work. We have provided point‑by‑point responses to your comments and would greatly appreciate it if you could take a moment to see whether we have addressed your concerns. We would be happy to engage further should you have any additional questions or suggestions.

---

> > > ### Comment · Reviewer_hq9t · 2025-08-05
> > >
> > > Thank you for your response and clarifications, which have addressed most of my concerns.

---

> > > > ### Author Response · Authors · 2025-08-05
> > > >
> > > > Thank you again for your thoughtful feedback and for engaging with our responses. If there are any remaining concerns that we could clarify or address, we would be very grateful for the opportunity to do so. We hope that resolving these points might allow you to consider adjusting your evaluation.

---

### Official Review · Reviewer_YkdX · 2025-07-04

**Clarity:** 2
**Significance:** 4
**Originality:** 3
**Rating:** 5
**Confidence:** 2

**Summary:**

This paper proposes a method for individual fairness in with respect to varying response costs: interventions on some feature(s), e.g. recourse. Specifically, the authors define their method wrt the Best Response Cost. The authors give significant analytical results over this problem, specifically focusing on randomized classifiers.

The proposed method is clearly outlined in Algorithms 1 and 2. The authors evaluate their method on two real-world datasets, FICO and Law School. Overall, the method evaluates very well, and flexibly trades off between F1 and individual fairness (alternatively: group fairness).

**Questions:**

Mostly from above:

1. Could you please elaborate on feature-attribution and and interpretability aspects of randomized classifiers? Is there any way to use a randomized classifier class that is both interpretable and individually-fair wrt intervention costs?

2. Is there a non-diferentiable heuristic for Ack which handles other base model classes?

**Ethical Concerns:**

["NO or VERY MINOR ethics concerns only"]

**Final Justification:**

This paper has strong analytical results on an interesting (and under-analyzed) class of model. The approach evaluates very well, showing controllable trade-offs through the hyperparameters.

The authors somewhat addressed my concern. But my first point is not a fatal flaw. Randomized model interpretability is out-of-scope of the paper (is the better answer), and future work gains in that model class are straightforwardly additive to this work.

**Limitations:**

Overall, the authors address the model's computational limitations, and the broader potential impacts. The authors demonstrate group- and individual-fairness, which is a strength in the fairness literature; the authors might consider more elaboration of the conflict between groupwise and individual fairness to highlight that this method doesn't sacrifice one for the other.

**Quality:**

3

**Strengths And Weaknesses:**

Strengths

1. Overall, this is a significant result in individual fairness under recourse. In several domains (e.g. financial), recourse is a legal requirement, and there is much concern about fairness implications (both group and individual) for these recourse responses.

2. The model seems to evaluate quite well, and the evaluation methodology is strong.

3. The model is quite flexible to both individual fairness and statistical parity regularization, L_c, Ω, respectively. This shows significant contribution to group fairness under recourse as well.

Weaknesses

1. The primary weakness seems to be the requirement for a randomized classifier class. While fair recourse is crucial in some domains (Strength #1), so is model *interpretability*. A randomized class of model seems less amenable to feature-attribution scores (e.g. with SHAP), and other governance requirements for these same domains. So in a sense, it seems one step forward (recourse), and one step back (governance).

2. The class of base classifier here seems very restrictive. Assumption 2.1 assumes continuous and differentiable. But tree-based models tend to be ubiquitous in the domains relevant to recourse (e.g. finance). So, overall, what is the problem domain where the base model is differentiable, *and* recourse is of value, *and* interpretability isn't?

While I think the result is significant in the IF literature, my doubt is whether the model addresses the paired limitations in strategic classification.

(Small) Clarity: In the limitations, the authors note that the linear optimization problem with K variables is 2^ K. But in the evaluation section, the binning factor is K=80, 100. This seems(?) intractable. Are these different uses of K?

Disclaimer: Some of the details around group fairness Eqs 18-22 I only understood intuitively. I defer the correctness of these to another reviewer. For these analytical results I'm primarily dropping my confidence.

---

> ### Author Rebuttal · Authors · 2025-07-31
>
> **Weakness 1 and Q1**: Our method implements a randomized classifier by sampling from a set of deterministic classifiers. For feature-attribution scores such as SHAP, we can compute them with respect to the expected prediction, $\mathbb{E}[\mathscr{F}(x)]$. While individual predictions may appear less interpretable due to the inherent randomness, the expected prediction across individuals with similar features remains interpretable and meaningful.
>
> From a governance perspective, the governing body can access both the randomly generated threshold and the resulting prediction to verify the correctness of any specific decision. Furthermore, it can monitor the distribution of thresholds $t$ over time to ensure that the randomization behaves as intended and aligns with fairness and performance requirements.
>
> **Weakness 2**: We do not assume that our base classifiers are differentiable. In Assumption 2.1, $l(x)$ is a score function, such as a default risk score commonly used in finance. Our actual base classifiers are defined as $f(x; t) = \mathbf{1}[l(\Delta x) \geq t]$ (see Eq. 3), which are not differentiable. Moreover, there is substantial literature on using regression models to do financial assessment [1, 2, 3]. The predicted risk scores from such models can naturally be viewed as differentiable functions $l(x)$.
>
> **Clarity**: The worst-case complexity of linear optimization algorithms can be as high as $2^{K}$. However, in practice, algorithms such as the simplex method typically solve problems in polynomial time for most cases.
>
> **Q2**: When $l(x)$ itself is not differentiable, e.g., a decision tree, we can apply randomization to every node of the tree. Achieving the optimal randomization policy needs to be considered with the specific application.
>
> **Limitations**: Thank you for your suggestion. We will elaborate more about the relationship between individual fairness and group fairness in the next version.
>
> [1] Kogan, Shimon, et al. "Predicting risk from financial reports with regression." Proceedings of human language technologies: the 2009 annual conference of the North American Chapter of the Association for Computational Linguistics. 2009.
>
> [2] Valaskova, Katarina, Tomas Kliestik, and Maria Kovacova. "Management of financial risks in Slovak enterprises using regression analysis." Oeconomia copernicana 9.1 (2018): 105-121.
>
> [3] Smita, Mrinalini. "Logistic regression model for predicting performance of S&P BSE30 company using IBM SPSS." International Journal of Mathematics Trends and Technology-IJMTT 67 (2021).

---

> > ### Comment · Reviewer_YkdX · 2025-08-01
> > **rebuttal acknowledgement**
> >
> > Dear authors,
> >
> > Thank you for your detailed rebuttal.
> >
> > 1. I'm not convinced that the randomized classifier class is equally interpretable. Reducing the randomized classifier to a discrete one using mean reduction is still a lossy aggregation; it's really unsatisfying for capturing the power of the randomized classifier (otherwise, why not simply use the mean classifier through the paper?)
> >
> > 2. My apologies. This was my misreading. Thank you for the clarification.
> >
> > I think that an Accept score is still fair; I do wish the authors all the luck for this review process. This work was a pleasure to read and, in my estimation, has a valuable contribution

---

> > > ### Author Response · Authors · 2025-08-01
> > >
> > > Thank you for the thoughtful follow-up and the Accept score. We agree with your opinion that interpretability for a randomized classifier is not as good as the deterministic one, although SHAP based on the expected outcome can provide some insights. We will also include a note about this point in the limitation section in the final version. Thanks again for your constructive comments.

---

### Note · Authors · 2025-08-13

We are grateful to the Reviewers and the Area Chair for their thoughtful engagement and constructive feedback throughout the review process. The discussion reinforced several aspects of our work that were viewed positively, including the novelty of framing individual fairness in strategic classification, our linear programming–based approach, and the integration of individual and group fairness objectives.

The central concern raised was that our assumption on the cost function, tied to $l(x)$, could restrict the broader applicability of our results. In our rebuttal, we addressed this by presenting a more general theorem that relaxes this assumption, thereby extending the scope and relevance of our framework. We believe this refinement strengthens the theoretical foundation of our work.

We appreciate the reviewers’ recognition of these clarifications and the constructive spirit of the exchange. Thank you for your work!

---

### Decision · Program_Chairs · 2025-09-17

**Decision:**

Accept (poster)

**Comment:**

The paper studies individual fairness in a strategic classification setting in which individuals may modify their features to influence machine learning decisions. All the reviewers are positive or mildly positive about the paper, and the rebuttal helped to clear out the reviewers' initial concerns.